# Burnout Among Healthcare Workers: Insights for Holistic Well-Being

**DOI:** 10.3390/healthcare13243298

**Published:** 2025-12-16

**Authors:** Carina Fernandes, Carla Barros, Pilar Baylina

**Affiliations:** 1Faculty of Human and Social Sciences, University Fernando Pessoa, 4249 Porto, Portugal; carinafernandes@ufp.edu.pt; 2School of Health, Polytechnic of Porto, 4200 Porto, Portugal

**Keywords:** burnout, psychosocial risk factors, healthcare workers, job demands, emotional strain

## Abstract

**Background/Objectives**: In the healthcare sector, burnout has become a critical concern due to high job demands and emotional strain. The main objective of the study is to examine the predictive role of psychosocial work-related risks in the development of burnout. **Methods**: A cross-sectional study was conducted, using a snowball recruitment method, from May to September 2025, among 154 healthcare workers. Data were collected using the psychosocial risk factors scale (INSAT_ERPS) and the Burnout Assessment Tool (BAT) and analyzed with descriptive and inferential statistics to analyze the predictive role of the psychosocial risk on burnout dimensions. **Results**: Psychosocial risk factors are consistently linked to the development of burnout symptoms. For exhaustion, the predictors included Working Hours (β = 0.312, *p* < 0.001), Social Work Relations (β = 0.196, *p* = 0.026), and Emotional Demands (β = 0.295, *p* = 0.002). For mental distance, the predictors included Work Intensity (β = −0.193, *p* = 0.049), Emotional Demands (β = 0.294, *p* = 0.004), and Work Values (β = 0.348, *p* = 0.003). For cognitive impairment, Work Values (β = 0.240, *p* = 0.042) and for emotional impairment, Employment Relations (β = 0.182, *p* = 0.038) emerged only one significant positive predictor. **Conclusions**: Findings underscore a crucial understanding: the development of burnout is not solely determined by the workload intensity, or the number of hours worked, the quality of working life and the dynamics within the workplace play pivotal roles in predicting burnout. A multidomain evaluation aligns with a holistic well-being approach to well-being by emphasizing that enhancing healthcare workers’ health demands systemic interventions addressing psychosocial work environment.

## 1. Introduction

Prolonged exposure to stressful and unfavorable working conditions tends to erode well-being. Over time, this sustained stress can culminate in burnout, when employees become emotionally and physically drained and disengaged from their work [1,2]. Burnout is a global issue characterized by a work-induced syndrome of chronic stress that has not been successfully managed [3]. In the 11th revision of the International Classification of Diseases of World Health Organization (ICD-11), burnout is defined as an occupational phenomenon with risk of harming health, comprising three factors: physical and emotional exhaustion, cynicism or mental distance (i.e., detachment and negativity toward work), and reduced professional efficacy [4]. When professionals experience burnout, they are in a state of physical and/or psychological exhaustion characterized by emotional fatigue, cold and dehumanized attitudes, behaving with cynicism and detachment, and experiencing feelings of incompetence and professional demotivation [5,6,7].

In the healthcare sector, burnout has become a critical concern due to high job demands and emotional strain. Even before the COVID-19 pandemic, burnout among healthcare workers was recognized as a serious issue, being the subject of urgent calls to action in healthcare systems [8]. Large-scale studies and reviews indicate that prior to 2020, about one in three clinicians experienced burnout symptoms. During the COVID-19 pandemic, this proportion increased dramatically, with a meta-analysis found that over half of healthcare workers (52%) met burnout criteria, with rates as high as 66% among physicians and nurses [9]. Between the causes, frontline providers such as physicians, nurses, and allied health professionals often report long working hours, heavy workloads, imbalances in duty allocation, physically strenuous work, resource constraints, and exposure to suffering [10]. Considering this evidence, burnout in healthcare is now recognized as a public and occupational health priority, demanding urgent attention and action [11].

At an individual level, workers with burnout often suffer from higher rates of insomnia, depression, anxiety, and psychosomatic complaints. Moreover, previous evidence has found that clinicians experiencing burnout are more prone to substance use [12] and even suicidal ideation [1]. At an organization level, the effects of burnout are equally alarming, as burnout can impair job performance and increase the likelihood of errors, threatening patient safety and quality of care. For instance, studies have linked higher burnout among physicians and nurses to more frequent medical mistakes and lower patient satisfaction [13,14,15]. Finally, at a system level, burnout drives higher absenteeism and turnover as exhausted clinicians are more likely to cut back their work hours or leave the profession, exacerbating workforce shortages [16,17,18,19]. This turnover incurs significant financial costs, as replacing an experienced nurse or physician is expensive and can reduce organizational productivity [20,21].

Psychosocial risks play an important role. Psychosocial risk factors at workplace include a set of dimensions of work environment that affects health and well-being that are less visible, like workload and time pressure, role ambiguity, social relations conflicts, emotional demands and lack of organizational support [22,23,24]. Thus, the quality of working life and the dynamics within the workplace play pivotal roles in predicting burnout; specifically, employment relations, emotional demands and ethical and value conflicts emerge as significant predictors of burnout, highlighting the psychological strain resulting from the misalignment between personal values and organizational demands leading to emotional exhaustion and professional disengagement [23,25,26,27]. Moreover, they validate the detrimental impact of poor employment relationships on psychological well-being, in line with the framework proposed by Maslach and Leiter [5] suggesting that professionals in fields where emotional labor is prevalent are particularly susceptible to burnout. In fact, psychosocial risk factors exert a significant influence on the manifestation of burnout symptoms, highlighting the importance of research in this field for understanding the widespread impact of these risks on professionals.

Thus, if unmitigated, the burnout crisis undermines healthcare delivery by diminishing care quality, increasing costs, and straining the capacity of health systems to meet patient needs. It is therefore unsurprising that burnout has been officially acknowledged as a work-related illness or occupational disorder in several European nations [28]. Moreover, the European Occupational Safety and Health Framework Directive (89/391/EEC-OSH) [29] requires employers to assess all potential risks to workers’ safety and health. To meet the requirements of this directive, organizations need to have access to valid and reliable methodologies capable of assessing employees’ levels of burnout [30].

Theoretical models of occupational stress, such as the Job Demands–Resources model, explain burnout as the outcome of chronically high job demands coupled with insufficient resources (e.g., lack of control or support) [31]. However, current research points to a range of psychosocial risk factors in the work environment as key contributors to burnout in healthcare settings. Among these stressors, we can find work process inefficiencies (e.g., computerized order entry and documentation), excessive workloads (e.g., work hours, overnight call frequency, nurse-patient ratios), work-home conflicts, organizational climate factors (e.g., management culture; lack of physician-nurse collaboration, value congruence, opportunities for advancement, and social support), and deterioration in control, autonomy, and meaning at work [32,33,34,35,36,37,38,39,40,41]. Moral distress, stemming from factors such as perceived powerlessness, unnecessary care, inadequate informed consent, and false hope [42], is also a significant predictor of burnout among nurses [36]. Recognizing these risk factors is crucial because they are modifiable. Theory and evidence together indicate that improving the psychosocial work environment can mitigate burnout and enhance well-being [43].

Given the multifaceted nature of burnout, a multidimensional assessment approach is essential to comprehensively capture its scope and underlying determinants. Accordingly, the present study employs a comprehensive set of validated instruments to evaluate work-related psychosocial risk factors and their predictive role in burnout among healthcare professionals.

This study encompasses multiple dimensions of burnout syndrome and different factors of the psychosocial work environment. Multidomain evaluation aligns with a holistic well-being perspective by recognizing that improving healthcare workers’ well-being requires addressing both the work environment and the individual’s health. Such a holistic assessment is not only diagnostic but also forms the foundation for intervention. The present study aims to explore the causes of burnout dimensions to suggest evidence-based strategies to improve healthcare workers’ well-being. Understanding exactly which psychosocial risk factors are most strongly related to burnout dimensions will enable targeted organizational changes and supportive interventions.

Therefore, this study aims to: (1) describe the burnout dimensions of the healthcare workers; (2) describe the levels of psychosocial risks to which healthcare workers are exposed; and (3) examine the predictive role of psychosocial work-related risks in the development of burnout.

## 2. Materials and Methods

### 2.1. Participants

This cross-sectional study employed a non-probabilistic convenience sampling approach, using a snowball recruitment method. Participants were recruited via posts shared on social media platforms (such as Facebook, version 542.0.0.47.151 and LinkedIn, version 4.1.1150) and through university students who agreed to distribute the study within their personal networks, from May to September 2025. Eligibility criteria included being a healthcare professional at least 18 years old and being fluent in European Portuguese.

### 2.2. Instruments

In addition to a socio-demographic questionnaire that included items on age, gender, educational level, and marital status, participants completed the Portuguese versions of the following instruments.

The psychosocial risk factors scale (INSAT-ERPS) is a Portuguese self-report questionnaire designed to assess working-related psychosocial risk factors. It includes 44 items organized into seven categories: work intensity (11 items; e.g., “Frequent interruptions”), working hours (6 items; e.g., “Exceeding normal working hours”), autonomy and initiative (4 items; e.g., “Not being able to participate in decisions regarding my work”), work social relations (8 items; e.g., “Needing help from colleagues and not receiving it”), employment relations (6 items; e.g., “I feel exploited most of the time”), emotional demands (5 items; e.g., “Being exposed to others’ difficulties and/or suffering”), and Work Values (4 items; e.g., “My professional conscience is undermined”) [22].

All items are rated on a 6-point Likert scale ranging from 0 (not exposed) to 5 (exposed with high discomfort). INSAT_ERPS has shown good internal consistency, with a reliability coefficient above 0.85 [22]. In a recent study composed by 356 participants, the INSAT Cronbach’s alpha coefficients were 0.972 for the total scale and, respectively, 0.909 for work intensity, 0.897 for working hours, 0.855 for autonomy and initiative, 0.814 for social work relations, 0.895 for employment relations, 0.941 for emotional demands, and 0.893 for work values [23]. Scores starting at 1.60 indicate a medium risk level, while scores above 3.30 indicate high risk [23]. Cronbach’s alpha values for this sample were 0.932 for total scale and for each category: 0.898 for work intensity, 0.879 for working times, 0.835 for autonomy and initiative, 0.915 for work social relations, 0.882 for employment relations with the organization, 0.919 for emotional demands, and 0.898 for work values.

Burnout Assessment Tool (BAT) [44]. The BAT is a self-report measure that conceptualizes burnout as a syndrome comprising four core dimensions—exhaustion, mental distance, emotional impairment, and cognitive impairment—assessed through 23 items. All items are rated on a five-point Likert scale ranging from 1 (“Never”) to 5 (“Always”) [45]. The four dimensions of the BAT evaluated: (a) Exhaustion (eight items, e.g., “At work, I feel mentally exhausted”), which reflects a severe depletion of energy resulting in both physical and mental fatigue. (b) Emotional impairment (five items, e.g., “At work, I feel unable to control my emotions”), which denotes intense emotional reactions and feelings of being overwhelmed by one’s emotions. (c) Cognitive impairment (five items, e.g., “At work, I have trouble staying focused”), which captures difficulties in memory, attention, and concentration, as well as reduced cognitive performance. (d) Mental distance (five items, e.g., “I struggle to find any enthusiasm for my work”), which reflects psychological detachment from work, often expressed as a strong reluctance or aversion to engaging in work-related activities [44]. Cronbach’s alpha values for this sample were 0.943 for all scale and for each category: 0.933 for exhaustion, 0.899 for mental distance, 0.844 for cognitive impairment, and 0.880 for emotional impairment.

### 2.3. Procedures

#### 2.3.1. Data Collection

This research followed a descriptive, observational, and cross-sectional quantitative design. Authorization to use the original instruments was formally obtained from their respective authors, and the study received approval from the local ethics committee (reference: FCHS/PI475/23). The questionnaire included various scales, starting with a cover page that briefly explained the study’s objectives. The criteria for participation involved informed consent, voluntary involvement, and confidentiality. Informed consent was obtained from all participants before they completed the survey. The estimated time to complete the questionnaire was about 15 min.

#### 2.3.2. Data Analysis

All statistical analyses were performed using IBM SPSS statistical program for Windows, version 30.0 (SPSS Inc., Chicago, IL, USA). A significance level of α = 0.05 was adopted. Descriptive statistics were computed for continuous variables, including the mean (M), standard deviation (SD), and range (minimum and maximum values), while categorical variables were summarized using absolute frequencies (n) and relative percentages (%). Pearson’s bivariate correlations were performed to examine the relationships among the INSAT_ERPS subscales (work-related psychosocial risks), and the BAT subscales (burnout dimensions). Correlations were interpreted according to Cohen’s guidelines, where *r* values of 0.10, 0.30, and 0.50 represent small, medium, and large effects, respectively [46].

To examine the predictive role of the INSAT_ERPS subscales on burnout dimensions, multiple linear regression analyses were conducted. Each core dimension of the BAT (exhaustion, mental distance, emotional impairment, and cognitive impairment) was entered as a dependent variable in separate regression models. The seven subscales (work intensity, working hours, autonomy and initiative, work social relations, employment relations, emotional demands, and work values) were simultaneously entered as independent predictors. Each model tested the extent to which the different facets of work-related psychosocial risks predicted specific burnout symptoms.

Model significance was evaluated using the *F*-test, and the amount of explained variance was expressed by the coefficient of determination (*R*^2^). The standardized regression coefficients (β) and their respective *p*-values were examined to identify significant predictors. Assumptions of linearity, homoscedasticity, and normality of residuals were verified. Multicollinearity was assessed through the Variance Inflation Factor (VIF), with values below 5 indicating acceptable levels of independence among predictors [47]. Only predictors with *p* < 0.05 were considered statistically significant.

## 3. Results

### 3.1. Sample

The sample was composed of 154 participants (M_age_ = 41.6; SD_age_ = 11.2; Age range = 21–67; 92.2% female). All participants were employed in the health care sector. Concerning their professional role, 43.5% were nurses, 22.1% were physicians, and 34.4% were other health professionals, including physiotherapists, psychologists, social workers, and allied health technicians. Regarding educational level, most participants held a university degree (59.1%) or a master’s degree (33.1%). A small number had completed secondary education (3.9%) or a doctoral degree (3.2%). In terms of marital status, 63.9% were married or in a civil partnership, 29.0% were single, and 6.5% were divorced or separated. Most participants worked in public institutions (61.3%), while 25.8% were employed in the private sector and 11.6% in public–private partnerships. Concerning employment type, 81.9% held a permanent contract, 7.7% were on a fixed-term contract, 9.0% worked under temporary or freelance arrangements, and 0.6% were self-employed.

### 3.2. Burnout Assessment Among Healthcare Professionals

The descriptive analysis of the BAT (mean scores) is presented in Table 1. Descriptive analysis of BAT scores (N = 154). The scale establishes two cut-off points: (a) scores beginning at 2.59 indicate burnout risk, and (b) scores exceeding 3.02 suggest positive burnout diagnoses [45]. As shown, and using these cut-off points, only the Exhaustion core symptom category indicates Burnout risk (M > 2.59) for all healthcare professionals.

Across all healthcare professionals, exhaustion is the predominant burden, with nurses appearing most affected overall (especially on exhaustion and emotional impairment). Physicians are like nurses in exhaustion but show greater variability in mental distance. Other staff report consistently lower burnout scores across dimensions.

### 3.3. Psychosocial Risk Factors Among Healthcare Professionals

Descriptive statistics for the INSAT_ERPS subscales are presented in Table 2. The results of the total sample indicate that, on average, participants reported medium levels of psychosocial risk across most categories. The highest mean scores were observed for Emotional Demands and Work Intensity, both above the medium-risk threshold. Work Values, Working Hours, and Employment Relations also fell within the medium-risk range. Conversely, Autonomy and Initiative and Social Work Relations reflected comparatively low levels of reported risk. When analyzing the results by professional category, physicians and nurses reported the highest average scores across most dimensions. Other health professionals generally presented lower mean scores across dimensions.

### 3.4. Predictive Role of Work-Related Psychosocial Risk on Burnout

Pearson’s correlations showed positive and statistically significant associations between Burnout dimensions (BAT) and INSAT_ERPS dimensions (*p* < 0.001).

For Exhaustion: correlations ranged from r = 0.307 to r = 0.561. The strongest were with Emotional Demands (r = 0.561), Work Values (r = 0.552), Working Hours (r = 0.523), and Social Work Relations (r = 0.492). A multiple linear regression examined how the seven INSAT dimensions predicted exhaustion. The model was significant, F(7146) = 16.51, *p* < 0.001, explaining 44.2% of variance (R^2^ = 0.442; adjusted R^2^ = 0.415). Durbin–Watson = 1.80 indicated independence of residuals. Significant predictors were Working Hours (β = 0.312, t = 3.60, *p* < 0.001), Social Work Relations (β = 0.196, t = 2.25, *p* = 0.026), and Emotional Demands (β = 0.295, t = 3.15, *p* = 0.002). The remaining predictors were not significant (*p* > 0.05). The results of the multiple linear regression analysis are presented in Table 3.

For Mental Distance: Correlations ranged from r = 0.220 to r = 0.534. The strongest were with Work Values (r = 0.534), Emotional Demands (r = 0.488), Autonomy and Initiative (r = 0.428), and Social Work Relations (r = 0.412). Multiple regression showed the model was significant, F(7146) = 11.65, *p* < 0.001, explaining 35.8% of variance (R^2^ = 0.358; adjusted R^2^ = 0.328). Durbin–Watson = 1.70 indicated independence of residuals. Significant predictors were Work Intensity (β = −0.193, t = −1.99, *p* = 0.049), Emotional Demands (β = 0.294, t = 2.93, *p* = 0.004), and Work Values (β = 0.348, t = 2.98, *p* = 0.003). It is noteworthy that work intensity appears as a potential protective factor, as indicated by its negative coefficient. Remaining predictors were not significant (*p* > 0.05). The results of the multiple linear regression analysis are presented in Table 4.

For Cognitive Impairment: Correlations ranged from r = 0.332 to r = 0.520. The strongest were with Work Values (r = 0.520), followed by Work Intensity (r = 0.486), Working Hours (r = 0.470), and Emotional Demands (r = 0.454). Multiple regression showed the model was significant, F(7146) = 11.48, *p* < 0.001, explaining 35.5% of variance (R^2^ = 0.355; adjusted R^2^ = 0.324). Durbin–Watson = 1.96 indicated independence of residuals. Work Values (β = 0.240, t = 2.05, *p* = 0.042) was the only significant predictor; all others were not significant (*p* > 0.05). The results of the multiple linear regression analysis are presented in Table 5.

For Emotional Impairment: Correlations ranged from r = 0.308 to r = 0.411. The strongest were with Social Work Relations (r = 0.411), Work Values (r = 0.396), Working Hours (r = 0.380), and Emotional Demands (r = 0.378).

Multiple regression showed the model was significant, F(7146) = 7.34, *p* < 0.001, explaining 26.0% of variance (R^2^ = 0.260; adjusted R^2^ = 0.225). Durbin–Watson = 1.60 indicated independence of residuals. Employment Relations (β = 0.182, t = 2.10, *p* = 0.038) was the only significant predictor; all others were not significant (*p* > 0.05). The results of the multiple linear regression analysis are presented in Table 6.

## 4. Discussion

Burnout is a reaction to ongoing workplace stress that can develop into a syndrome marked by depersonalization, diminished sense of personal accomplishment, and emotional tiredness. [5,48]. In fact, healthcare workers who experienced burnout are in a state of physical and/or psychological tiredness that is characterized by emotional exhausting, dehumanizing and cold attitudes, cynical and detached behavior, feelings of incompetence, and demotivation in the workplace. The psychological effects of burnout syndrome include deterioration of cognitive, emotional, and attitude traits, as well as antagonistic actions towards one’s professional identity, coworkers, clients, and workplace [6,7,26,49].

Exploring psychosocial risks is essential for understanding and mitigating burnout in the workplace. These risks, such as excessive workload, poor social support, and emotional demands, are consistently linked to the development of burnout symptoms, including exhaustion, mental distance, cognitive impairment, and emotional impairment [8,43].

Exhaustion, recognized as a central dimension of burnout, is strongly influenced by a combination of organizational and interpersonal workplace factors. The results showed that the most robust predictors of exhaustion were working hours (β = 0.312; *p* < 0.001) and emotional demands (β = 0.295; *p* = 0.002), while work social relations also contributed significantly (β = 0.196; *p* = 0.026). Our results shows that the model explain the high variance of the predictors (44.2%) These findings underscore the multifaceted nature of exhaustion, which arises not only from workload intensity but also from the emotional and social relations in the workplace. These predictors justify the complexity of work activity, showing the difficult relationship between organizational demands, interpersonal dynamics, and emotional strain faced by healthcare professionals.

Prolonged working hours have consistently been linked to higher exhaustion levels. Healthcare professionals working more than 40 h per week—often under irregular or extended schedules—report significantly higher burnout scores, particularly in the domain of emotional exhaustion [24,50]. This situation limits the chances for psychological detachment and recovery, two key mechanisms for maintaining wellbeing. In healthcare settings, long hours are frequently compounded by emotional demands, which refer to the sustained psychological effort required to manage one’s emotions during patient interactions. These demands include carrying hard news, maintaining behavior in high-stress situations, and expressing empathy under pressure lead to deeper physical and emotional fatigue, especially when emotional regulation is performed without adequate support [27,51]. In fact, poor work social relations—characterized by low interpersonal trust, lack of collaboration, and support—can exacerbate exhaustion. When employees feel isolated or undervalued, their capacity to cope with emotional and workload stressors diminishes. Conceição and Palma-Moreira [25] emphasize that rigid work environment and poor interpersonal dynamics heighten emotional strain, making employees more vulnerable to burnout. Similarly, Wekenborg et al. [52] found that individuals experiencing burnout show impaired social decision-making and reduced prosocial behavior, which undermines team cohesion and weakens collective support.

Together, these factors interact in a complex way to cause fatigue in healthcare settings: long hours can amplify the impact of emotional demands, especially when social support is weak. In fact, poor social relations reduce resilience and increase emotional dysregulation, accelerating exhaustion [53]. In addition, making organizational adjustments to workload and scheduling, addresses to a more focused interventions to improve relationships at work and provide emotional support, and promote wellbeing.

Mental distance is a core dimension of burnout characterized by a sense of alienation from one’s work, a loss of enthusiasm, and a psychological detachment from it. Our results showed that the strongest predictors of mental distance were work values (β = 0.348; *p* = 0.003) and emotional demands (β = 0.294; *p* = 0.004).

Work values emerge as significant predictors of burnout, highlighting the psychological strain resulting from the misalignment between personal values and organizational demands. When their work lacks personal purpose or conflicts with their own ideals, especially in emotionally demanding professions, this discrepancy causes an emotional retreat and a hard internal conflict. These feelings of disengagement emerges when organizational demands are in conflict with ethical or personal values of the worker; and can be seen as a self-defense strategy, as a form of self-protective emotional detachment [34]. This burnout dimension is very relevant and can have hard consequences on healthcare workers wellbeing. Typically accompanied by high emotional demands, mental distance can have profound consequences on both individual health and organizational effectiveness, leading to reduced empathy, impaired decision-making, and diminished quality of care [51]. In fact, this detachment is closely linked to cynicism, a burnout dimension characterized by negative attitudes toward one’s job, colleagues or organization, but is a form of psychological withdrawal, where employees emotionally disconnect from their work to cope with stressors [54].

In the context of burnout, cognitive impairment embraces issues linked to cognitive and executive functioning, namely memory, attention, and decision-making. It is a fundamental dimension of burnout that impacts both quality of life and professional effectiveness. In our study cognitive impairment was linked to work values (β = 0.240; *p* = 0.042). This psychosocial risk factor is a significant predictor of burnout that can be found when employees are compelled to work in settings where the organizational culture contradicts their core work values, leading to internal conflict, and a gradual deterioration of professional identity.

Also found in mental distance, this internal conflict can exacerbate these cognitive impairments by increasing psychological strain, reducing cognitive resources and cognitive decline (Renaud & Lacroix [55]; Koutsimani et al. [56]. In a systematic review, Gavelin et al. [48] came to the conclusion that psychosocial risk factors had a significant impact on cognitive functioning. They found that cognitive tiredness and decreased mental flexibility were significantly predicted by limited job control and poor value alignment. In healthcare workers we can find that work values incongruence between healthcare workers and institutional priorities, such as giving efficiency precedence over patient care and safety, contributes significantly to emotional exhaustion, inability to make decisions and depersonalization [57].

Emotional impairment in burnout refers to diminished emotional regulation, increased irritability, and difficulty managing interpersonal interactions. Particularly in occupations requiring intense emotional interactions, such as healthcare workers, it frequently shows up as emotional tiredness, detachment, and less empathy. In our study, emotional impairment was linked to employment relations (β = 0.182; *p* = 0.038), that includes organizational procedures and norms, leadership management that characterized the culture of work context. In fact, mechanisms of employment relations like rigid or unsupportive employment structures, lack autonomy or low recognition and trust, increases vulnerability that can lead to emotional dysregulation, chronic emotional strain and burnout cynicism behaviors Chen et al. [51]; Conceição & Palma-Moreira [25]. Moreover, they validate the detrimental impact of poor employment relationships on emotional and psychological well-being, in line with the framework proposed by Maslach and Leiter [5].

Findings underscore a crucial understanding: the development of burnout is not solely determined by the workload intensity or the number of hours worked. Instead, this study reveals that the quality of working life and the dynamics within the workplace play pivotal roles in predicting burnout.

Moreover, they are consistent with the Job Demands–Resources Theory: disengagement and cognitive/emotional functioning are shaped by person–job mismatch and organizational fairness, respectively, while quantitative and emotional demands exhaust resources [31]. In fact, burnout is not just an individual issue but a systemic one. Poor organizational culture—marked by distrust, lack of recognition, and inadequate communication—was linked to emotional detachment and reduced empathy among healthcare staff [57].

Evidence suggests that interventions can indeed make a positive difference. According to a recent systematic review of intervention studies [58], initiatives aimed at enhancing the well-being of healthcare workers frequently resulted in notable decreases in staff levels of burnout, stress, anxiety, and depression as well as notable improvements in resilience, work engagement, well-being, and quality of life. However, the majority of interventions were secondary-level (helping individuals manage stress and other individual strategies), and a smaller number were primary-level interventions (proactive steps by organizations to eliminate sources of stress, such as workflow changes or better staffing) [59]. This suggests that while we have a toolkit of potentially helpful strategies, more rigorous research and comprehensive programs are needed to fully address the burnout crisis, centered on work activity and psychosocial risk factors, through a holistic well-being approach.

In fact, work organization influences the level of burnout of healthcare workers: task-related risk factors include not having enough time to finish a task, workflow interruptions happening too often, not having enough or clear information, and contradictory demands between strict deadlines and the requirement to maintain high quality [8,10,60]. However, work environment characteristics, such social work relations, employment relations and emotional demands significantly impact burnout levels. Difficult interpersonal relationships, insufficient organizational support, and ongoing emotional stress all contribute to higher psychological distress and more vulnerable to burnout symptoms among healthcare workers.

Although this study offers important contributions, it is important to acknowledge the limitations. This study’s first limitation is related to the sample’s size, which is relatively small. Also, it presents a very different number of nurses and medical doctors, and male and female participants. As such, the obtained results cannot be solidly generalized for the population and group comparisons are hampered. This cross-sectional design with a self-reported questionnaire may introduce recall bias, like subjective perception of context factors like work environment and employment relations at the time of data collection, despite using validated instruments.

Despite these limitations, the study provides valuable evidence on burnout causes and can be expanded to enlarge the sample. Future studies, with a larger and more repre-sentative sample of the healthcare professionals would help confirm the validity of the findings. In this sense, it is also important to recognize this study contribute to the visibility of this phenomenon, both for the community, in general, and for healthcare professionals, in particular.

## 5. Conclusions

This work contributes to the expanding body of knowledge in occupational health by highlighting the important impact of psychosocial risk factors on the onset of burnout and outlines the primary predictive determinants of burnout. It is essential to explore intervention strategies that companies can implement to mitigate these risks and underscores the importance of proactive planning approach. From the individual to the organizational level, strategies should aim to foster a work environment that supports professional fulfillment and overall well-being. This includes implementing psychological support programs, promoting strong social networks, reinforcing positive employment relations, encouraging peer collaboration, and humanizing supportive leadership. At the policy level, it is essential to strengthen occupational health plans through integrated evaluation and intervention, ensuring their effective implementation due to contextual barriers in healthcare systems. Finally, a comprehensive approach is required to address burnout, prioritizing organizational changes that safeguard psychological health and well-being.

## Figures and Tables

**Table 1 healthcare-13-03298-t001:** Descriptive analysis of BAT scores (N = 154).

BAT	Total Sample M (SD)	Physicians M (SD)	Nurses M (SD)	Other M (SD)
Exhaustion	2.98 (0.83)	3.02 (0.94)	3.10 (0.78)	2.79 (0.79)
Mental distance	1.95 (0.86)	2.05 (1.09)	1.99 (0.82)	1.82 (0.73)
Cognitive impairment	1.92 (0.78)	1.98 (0.85)	1.99 (0.81)	1.76 (0.67)
Emotional impairment	2.00 (0.64)	1.95 (0.75)	2.14 (0.74)	1.85 (0.69)

M—Mean; SD—Standard Deviation; Total sample (*n* = 154); Physicians (*n* = 36); Nurses (*n* = 67); Other health Professionals (*n* = 51).

**Table 2 healthcare-13-03298-t002:** Descriptive statistics for the INSAT_ERPS dimensions.

INSAT_ERPS	Total Sample M (SD)	Risk Level	Physicians M (SD)	Risk Level	Nurses M (SD)	Risk Level	Other M (SD)	Risk Level
Work Intensity	2.07 (0.97)	Medium	2.35 (0.92)	Medium	2.20 (0.88)	Medium	1.74 (1.03)	Medium
Working Hours	1.81 (1.09)	Medium	2.06 (1.21)	Medium	1.97 (0.99)	Medium	1.46 (1.07)	Low
Autonomy and Initiative	1.08 (1.04)	Low	1.43 (1.16)	Low	1.07 (0.91)	Low	0.87 (1.06)	Low
Social Work Relations	0.68 (0.76)	Low	0.67 (0.90)	Low	0.68 (0.67)	Low	0.68 (0.79)	Low
Employment Relations	1.76 (0.97)	Medium	1.59 (0.99)	Medium	1.91 (0.98)	Medium	1.69 (0.93)	Medium
Emotional Demands	2.18 (1.16)	Medium	2.34 (1.23)	Medium	2.43 (1.14)	Medium	1.77 (1.03)	Medium
Work Values	1.96 (1.39)	Medium	2.52 (1.48)	Medium	2.04 (1.40)	Medium	1.50 (1.18)	Low

M—Mean; SD—Standard Deviation; Scores of INSAT_ERPS below 1.60 reflect low psychosocial risk, between 1.60 and 3.29 indicate medium risk, while scores ≥ 3.30 indicate high risk [23]. Total sample (*n* = 154); Physicians (*n* = 36); Nurses (*n* = 67); Other health Professionals (*n* = 51).

**Table 3 healthcare-13-03298-t003:** Multiple Linear Regression Coefficients for the Prediction of the core symptom of Burnout—Exhaustion.

Predictor	*B*	*SE B*	*β*	*t*	*p*	95% CI [LL, UL]
(Constant)	2.052	0.141	**–**	14.55	<0.001	[1.774, 2.331]
Work intensity	−0.098	0.077	−0.114	−1.26	0.209	[−0.250, 0.055]
**Working hours**	**0.237**	**0.066**	**0.312**	**3.60**	**<0.001**	**[0.107, 0.367]**
Autonomy and initiative	−0.066	0.073	−0.082	−0.91	0.366	[−0.210, 0.078]
**Social work relations**	**0.213**	**0.095**	**0.196**	**2.25**	**0.026**	**[0.026, 0.401]**
Employment relations	−0.041	0.064	−0.048	−0.63	0.529	[−0.168, 0.087]
**Emotional demands**	**0.211**	**0.067**	**0.295**	**3.15**	**0.002**	**[0.079, 0.344]**
Work Values	0.120	0.065	0.201	1.85	0.066	[−0.008, 0.248]

Values in bold are statistically significant (*p* < 0.05).

**Table 4 healthcare-13-03298-t004:** Multiple Linear Regression Coefficients for the Prediction of the core symptom of Burnout—Mental distance.

Predictor	*B*	*SE B*	*β*	*t*	*p*	95% CI [LL, UL]
(Constant)	1.334	0.157	–	8.49	<0.001	[1.024, 1.645]
**Work intensity**	**−0.171**	**0.086**	**−0.193**	**−1.99**	**0.049**	**[−0.341, −0.001]**
Working hours	−0.064	0.073	−0.082	−0.88	0.382	[−0.210, 0.081]
Autonomy and initiative	0.124	0.081	0.149	1.53	0.127	[−0.036, 0.284]
Social work relations	0.101	0.106	0.089	0.96	0.340	[−0.108, 0.310]
Employment relations	−0.007	0.072	−0.008	−0.10	0.921	[−0.149, 0.135]
**Emotional demands**	**0.218**	**0.075**	**0.294**	**2.93**	**0.004**	**[0.071, 0.366]**
**Work Values**	**0.216**	**0.072**	**0.348**	**2.98**	**0.003**	**[0.073, 0.358]**

Values in bold are statistically significant (*p* < 0.05).

**Table 5 healthcare-13-03298-t005:** Multiple Linear Regression Coefficients for the Prediction of the core symptom of Burnout—Cognitive impairment.

Predictor	*B*	*SE B*	*β*	*t*	*p*	95% CI [LL, UL]
(Constant)	0.977	0.142	–	6.87	<0.001	[0.696, 1.258]
Work intensity	0.115	0.078	0.144	1.48	0.141	[−0.039, 0.269]
Working hours	0.114	0.066	0.161	1.72	0.087	[−0.017, 0.246]
Autonomy and initiative	−0.035	0.073	−0.046	−0.47	0.637	[−0.180, 0.110]
Social work relations	0.125	0.096	0.122	1.31	0.194	[−0.064, 0.314]
Employment relations	0.065	0.065	0.081	1.00	0.321	[−0.064, 0.193]
Emotional demands	0.032	0.068	0.048	0.47	0.636	[−0.102, 0.166]
**Work Values**	**0.134**	**0.065**	**0.240**	**2.05**	**0.042**	**[0.005, 0.263]**

Values in bold are statistically significant (*p* < 0.05).

**Table 6 healthcare-13-03298-t006:** Multiple Linear Regression Coefficients for the Prediction of the core symptom of Burnout—Emotional impairment.

Predictor	*B*	*SE B*	*β*	*t*	*p*	95% CI [LL, UL]
(Constant)	1.318	0.144	–	9.15	<0.001	[1.033, 1.603]
Work intensity	−0.049	0.079	−0.064	−0.62	0.538	[−0.205, 0.107]
Working hours	0.101	0.067	0.150	1.50	0.136	[−0.032, 0.234]
Autonomy and initiative	0.050	0.074	0.070	0.67	0.502	[−0.097, 0.197]
Social work relations	0.144	0.097	0.149	1.49	0.139	[−0.047, 0.335]
**Employment relations**	**0.138**	**0.066**	**0.182**	**2.10**	**0.038**	**[0.008, 0.268]**
Emotional demands	0.043	0.068	0.068	0.63	0.532	[−0.092, 0.178]
Work Values	0.059	0.066	0.112	0.89	0.373	[−0.072, 0.190]

Values in bold are statistically significant (*p* < 0.05).

## Data Availability

The data presented in this study are available on request from the corresponding author due to privacy reasons.

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
