# Peer review of "Burnout Among Healthcare Workers: Insights for Holistic Well-Being"

_healthcare, 2025, doi:10.3390/healthcare13243298_

Round 1
Reviewer 1 Report
Comments and Suggestions for Authors
Manuscript Review
The manuscript, Burnout Among Healthcare Workers: From Evaluation to Intervention for Holistic Well-Being, presents a well-designed and timely study that addresses the highly relevant and frequently explored, yet still insufficiently resolved, issue of burnout among healthcare workers. The introduction is comprehensive, well-structured, and clearly situates the phenomenon within the contemporary healthcare context, drawing on relevant and up-to-date sources.
One notable strength of this manuscript is the use of two modern, validated instruments to assess psychosocial risks and burnout: INSAT-ERPS and the Burnout Assessment Tool. This selection enhances the study’s methodological correctness, and it is worth emphasising that the BAT, in particular, has gained substantial recognition in recent years for its conceptual precision and multidimensional approach to measuring burnout. The statistical analyses are well planned and clearly presented, offering insight not only into correlations but also into specific predictive models linking psychosocial risks to different aspects of burnout. In this way, the authors provide a deeper understanding of the phenomenon and highlight how various job demands and value-related conflicts within the work environment contribute to emotional exhaustion, mental distancing, and cognitive difficulties among healthcare professionals.
Although the results are adequately presented, the methodological section appears insufficiently elaborated in certain sections. The recruitment of participants using the snowball sampling method is described briefly, with insufficient clarification of how the initial group was formed. Additionally, although the sample size of 154 participants is useful for generating preliminary insights, it remains relatively small and heterogeneous, considering the range of professional categories included.
The manuscript also lacks a clearly delineated section addressing the study’s limitations. Given the sampling method, sample size, heterogeneity among professional groups, and the inherent constraints of self-report measures, the authors should include a dedicated section discussing these limitations.
The abstract requires a minor yet important correction. The current statement about the use of “two validated questionnaires” is overly general and does not enhance methodological clarity. It is advisable for the authors to explicitly name the instruments INSAT-ERPS and BAT so readers are immediately informed of the methodological framework. Likewise, the expression “the main results show” should be replaced with a more neutral, scientifically appropriate phrasing.
Despite these minor revisions, it is important to highlight that the authors rely on an extensive, current, and highly relevant body of literature, including recent systematic reviews and empirical studies. This demonstrates that the research is firmly grounded in contemporary scientific evidence and provides a meaningful contribution to understanding psychosocial determinants of burnout.
Overall, the study is clearly written, and the results are presented in a way that is both informative and valuable for researchers and practitioners in healthcare. With the suggested minor adjustments, the manuscript is fully suitable for publication in Healthcare, and I therefore recommend minor revision.
Author Response
Dear Reviewer,
Thank you very much for taking the time to review this manuscript. Please find the detailed responses below and the corresponding revisions/corrections highlighted/in track changes in the re-submitted files.
The manuscript, Burnout Among Healthcare Workers: From Evaluation to Intervention for Holistic Well-Being, presents a well-designed and timely study that addresses the highly relevant and frequently explored, yet still insufficiently resolved, issue of burnout among healthcare workers. The introduction is comprehensive, well-structured, and clearly situates the phenomenon within the contemporary healthcare context, drawing on relevant and up-to-date sources.
One notable strength of this manuscript is the use of two modern, validated instruments to assess psychosocial risks and burnout: INSAT-ERPS and the Burnout Assessment Tool. This selection enhances the study’s methodological correctness, and it is worth emphasising that the BAT, in particular, has gained substantial recognition in recent years for its conceptual precision and multidimensional approach to measuring burnout. The statistical analyses are well planned and clearly presented, offering insight not only into correlations but also into specific predictive models linking psychosocial risks to different aspects of burnout. In this way, the authors provide a deeper understanding of the phenomenon and highlight how various job demands and value-related conflicts within the work environment contribute to emotional exhaustion, mental distancing, and cognitive difficulties among healthcare professionals.
R1. Although the results are adequately presented, the methodological section appears insufficiently elaborated in certain sections. The recruitment of participants using the snowball sampling method is described briefly, with insufficient clarification of how the initial group was formed. Additionally, although the sample size of 154 participants is useful for generating preliminary insights, it remains relatively small and heterogeneous, considering the range of professional categories included.
R1. We thank the reviewer for highlighting this concern. We improved the section Materials and Methods, Data collection, according to this comment. We fully agree about the sample size and we introduced a limitation section in the end of our Discussion Section. Please see the changes resulting from this comment on lines 454–467.
R2. The manuscript also lacks a clearly delineated section addressing the study’s limitations. Given the sampling method, sample size, heterogeneity among professional groups, and the inherent constraints of self-report measures, the authors should include a dedicated section discussing these limitations.
R2. We thank the reviewer for highlighting this concern. We fully agree and introduced a limitation section in the end of our Discussion Section with the constraints of our study. Please see the changes resulting from this comment on lines 454–467.
R3. The abstract requires a minor yet important correction. The current statement about the use of “two validated questionnaires” is overly general and does not enhance methodological clarity. It is advisable for the authors to explicitly name the instruments INSAT-ERPS and BAT so readers are immediately informed of the methodological framework. Likewise, the expression “the main results show” should be replaced with a more neutral, scientifically appropriate phrasing.
R3. We improved the abstract according to this comment to enhance the methodological clarity. Please see the changes on lines 14-19.
Despite these minor revisions, it is important to highlight that the authors rely on an extensive, current, and highly relevant body of literature, including recent systematic reviews and empirical studies. This demonstrates that the research is firmly grounded in contemporary scientific evidence and provides a meaningful contribution to understanding psychosocial determinants of burnout.
Overall, the study is clearly written, and the results are presented in a way that is both informative and valuable for researchers and practitioners in healthcare. With the suggested minor adjustments, the manuscript is fully suitable for publication in Healthcare, and I therefore recommend minor revision.

Reviewer 2 Report
Comments and Suggestions for Authors
This manuscript addresses an important and timely issue—burnout among healthcare workers—and examines the predictive role of psychosocial risk factors using validated instruments. The study is conceptually grounded, methodologically sound, and generally well written. It provides valuable empirical insights with practical implications for occupational health and organizational interventions.
However, several areas require clarification, refinement, and strengthening to enhance the rigor, coherence, and overall contribution of the manuscript. Below are my comments.
- The introduction could better articulate how this study differs from previous research, beyond stating that burnout is multifaceted. Explicitly identifying gaps in the literature, especially regarding the Portuguese context or the integration of psychosocial risk predictors, would strengthen the rationale. Therefore, clarifying the novelty of the study (e.g., limited studies linking specific INSAT dimensions to BAT outcomes; few holistic predictive models applied to diverse healthcare professions).
- In the instrument section, while both tools are described in the paper, the text becomes excessively lengthy, especially the BAT subscales description. This reads like a manual. Summaries might suffice, with details placed in supplementary files.
-
Study period is listed as May–September 2025, but no contextual information is provided (e.g., post-pandemic setting, relevant national healthcare conditions). Suggest to add more reflexivity on limitations and contextual factors.
- Several parts of the Results section repeat correlation findings before regression findings, making the narrative redundant. Suggest removing redundant information.
- The regression models explain variance ranging from 26% to 44%, which is meaningful. However, the discussion should explicitly highlight the practical significance and relative importance of specific predictors.
- A few predictors show negative beta values (e.g., work intensity predicting mental distance negatively). This requires explanation or at least an acknowledgement.
- While the discussion is thorough and well-referenced, it could still benefit from explicitly discussing how each finding confirms, contradicts, or extends prior studies. Also, reduce descriptive explanations of burnout concepts.
-
The manuscript concludes that burnout is systemic and requires organization-level interventions. This could be strengthened by providing more concrete examples of primary-level interventions, distinguishing between individual, organizational, and policy-level actions, and addressing feasibility and contextual barriers in healthcare systems.
- Some paragraphs are very long and could be broken up for readability.
- Minor grammatical issues (e.g., verb tenses, missing articles).
- Occasional misplaced commas and run-on sentences.
Author Response
Dear Reviewer,
Thank you very much for taking the time to review this manuscript. Please find the detailed responses below and the corresponding revisions/corrections highlighted/in track changes in the re-submitted files.
This manuscript addresses an important and timely issue—burnout among healthcare workers—and examines the predictive role of psychosocial risk factors using validated instruments. The study is conceptually grounded, methodologically sound, and generally well written. It provides valuable empirical insights with practical implications for occupational health and organizational interventions.
However, several areas require clarification, refinement, and strengthening to enhance the rigor, coherence, and overall contribution of the manuscript. Below are my comments.
- The introduction could better articulate how this study differs from previous research, beyond stating that burnout is multifaceted. Explicitly identifying gaps in the literature, especially regarding the Portuguese context or the integration of psychosocial risk predictors, would strengthen the rationale. Therefore, clarifying the novelty of the study (e.g., limited studies linking specific INSAT dimensions to BAT outcomes; few holistic predictive models applied to diverse healthcare professions).
R1. We thank the reviewer for highlighting this concern. We introduced new paragraph with the integration of psychosocial risk predictors of burnout to strengthen the rationale. Please see the changes resulting from this comment on lines 71–88.
- In the instrument section, while both tools are described in the paper, the text becomes excessively lengthy, especially the BAT subscales description. This reads like a manual. Summaries might suffice, with details placed in supplementary files.
R2. We appreciate the reviewer’s attention to this concern. We reviewed and improved the text. Please see changes on lines 164-179.
- Study period is listed as May–September 2025, but no contextual information is provided (e.g., post-pandemic setting, relevant national healthcare conditions). Suggest to add more reflexivity on limitations and contextual factors.
R3. Thank you for your valuable comment. We introduced this acknowledge in our limitations section. Please see the changes on lines 444–451.
- Several parts of the Results section repeat correlation findings before regression findings, making the narrative redundant. Suggest removing redundant information.
R4. We thank the reviewer for this comment. We fully agree and removed the redundant information in subsection 3.3. Please see the changes on lines 263-313.
- The regression models explain variance ranging from 26% to 44%, which is meaningful. However, the discussion should explicitly highlight the practical significance and relative importance of specific predictors.
R5. We thank the reviewer for this comment. Suggested information has been included. Please see changes on lines 335-340.
- A few predictors show negative beta values (e.g., work intensity predicting mental distance negatively). This requires explanation or at least an acknowledgement.
R6. We thank the reviewer for highlighting this concern. We fully agree and we improved the text. Please see the changes on lines 285-286.
- While the discussion is thorough and well-referenced, it could still benefit from explicitly discussing how each finding confirms, contradicts, or extends prior studies. Also, reduce descriptive explanations of burnout concepts.
R7. We thank the reviewer for highlighting this concern. We highlighted some studies that confirm our results. Please see changes on lines 354-358, 377-383, 394-397, 408-411.
- The manuscript concludes that burnout is systemic and requires organization-level interventions. This could be strengthened by providing more concrete examples of primary-level interventions, distinguishing between individual, organizational, and policy-level actions, and addressing feasibility and contextual barriers in healthcare systems.
R8. We thank the reviewer for this thoughtful comment. We fully agree and improved the conclusions text. Please see on lines 466-475.

Reviewer 3 Report
Comments and Suggestions for Authors
Dear Authors,
Your manuscript titled “Burnout among healthcare workers: from evaluation to intervention for holistic well-being” addresses an important question regarding how various psychological risk factors in the workplace shape employees’ burnout experiences. Your genuine care for the topic is clearly reflected in the writing. At the same time, I have several observations that I hope will be helpful as you refine the manuscript.
- Conceptual Definitions
In empirical papers, it is standard to introduce and define key variables when they first appear in the manuscript—typically in the introduction or the hypothesis development section, and certainly before the methodology section. In your current draft, the primary constructs—burnout and psychological risk factors, both of which are multidimensional—are not defined until the method section. This significantly limits the clarity of the research question and weakens the articulation of the study’s contributions. Early conceptual grounding would improve readability and overall merit of the study.
- Hypothesis Development
For quantitative studies, hypotheses should be clearly articulated prior to data collection, with explicit statements about the expected relationships between variables (e.g., positive or negative associations). These expectations should be grounded in relevant theoretical frameworks. In the current manuscript, the hypotheses are vague and lack explicit directional predictions. This is particularly problematic since you appear to analyze associations at the dimensional level between multiple psychological risk factors and the various dimensions of burnout.
Moreover, the manuscript does not offer theoretical reasoning linking each psychological risk factor to each burnout dimension. The relationships are often stated as factual rather than theoretically argued. This lack of theoretical justification weakens the logic of the model and diminishes the overall contribution. Strengthening the conceptual rationale would markedly improve the rigor of the study.
- Labeling of Psychological Risk Factors
The six psychological risk factors appear to lack a consistent labeling logic. Several are framed as positive constructs (e.g., “work social relations,” “employee relations,” “work values”) even though the manuscript’s descriptions suggest deficits or the absence of these qualities. This inconsistency creates conceptual ambiguity. The manuscript would benefit from clearer terminology that accurately reflects the nature of each factor and avoids contradictory framing.
- Paper Title
The title—“from evaluation to intervention for holistic well-being”—does not accurately represent the scope of the empirical work. The core focus of the study is the examination of contributors to burnout rather than holistic well-being, and the manuscript does not empirically explore intervention strategies. While holistic well-being and intervention may be addressed in practical implications, they are not the focus of the empirical investigation. The title should reflect the central research question rather than the applied implications.
I hope these comments are useful as you continue developing this important work.
Author Response
Dear Reviewer,
Thank you very much for taking the time to review this manuscript. Please find the detailed responses below and the corresponding revisions/corrections highlighted/in track changes in the re-submitted files.
Your manuscript titled “Burnout among healthcare workers: from evaluation to intervention for holistic well-being” addresses an important question regarding how various psychological risk factors in the workplace shape employees’ burnout experiences. Your genuine care for the topic is clearly reflected in the writing. At the same time, I have several observations that I hope will be helpful as you refine the manuscript.
- Conceptual Definitions
In empirical papers, it is standard to introduce and define key variables when they first appear in the manuscript—typically in the introduction or the hypothesis development section, and certainly before the methodology section. In your current draft, the primary constructs—burnout and psychological risk factors, both of which are multidimensional—are not defined until the method section. This significantly limits the clarity of the research question and weakens the articulation of the study’s contributions. Early conceptual grounding would improve readability and overall merit of the study.
R1. We thank the reviewer for highlighting this concern. We included in the Introduction Section a more clarify and articulate definitions of the primary constructs – burnout and psychosocial risk factors. Please see the changes resulting from this comment on lines 38–46 and 71-88.
- Hypothesis Development
For quantitative studies, hypotheses should be clearly articulated prior to data collection, with explicit statements about the expected relationships between variables (e.g., positive or negative associations). These expectations should be grounded in relevant theoretical frameworks. In the current manuscript, the hypotheses are vague and lack explicit directional predictions. This is particularly problematic since you appear to analyze associations at the dimensional level between multiple psychological risk factors and the various dimensions of burnout.
Moreover, the manuscript does not offer theoretical reasoning linking each psychological risk factor to each burnout dimension. The relationships are often stated as factual rather than theoretically argued. This lack of theoretical justification weakens the logic of the model and diminishes the overall contribution. Strengthening the conceptual rationale would markedly improve the rigor of the study.
R2. We thank the reviewer for highlighting this concern. We agree and we improve the Introduction text. Please see changes on lines 117-125.
- Labeling of Psychological Risk Factors
The six psychological risk factors appear to lack a consistent labeling logic. Several are framed as positive constructs (e.g., “work social relations,” “employee relations,” “work values”) even though the manuscript’s descriptions suggest deficits or the absence of these qualities. This inconsistency creates conceptual ambiguity. The manuscript would benefit from clearer terminology that accurately reflects the nature of each factor and avoids contradictory framing.
R3. We thank the reviewer for highlighting this concern. We Agree and we improve all text. The terminology is the same used in the validation article (Barros, C.; Cunha, L.; Rocha, A.; Baylina, P. The psychosocial risk factors scale: factorial validity and reliability analysis. Int. J. Occup. Saf. Ergon. 2025, 10.1080/10803548.2025.2566584, 1-15).
- Paper Title
The title—“from evaluation to intervention for holistic well-being”—does not accurately represent the scope of the empirical work. The core focus of the study is the examination of contributors to burnout rather than holistic well-being, and the manuscript does not empirically explore intervention strategies. While holistic well-being and intervention may be addressed in practical implications, they are not the focus of the empirical investigation. The title should reflect the central research question rather than the applied implications.
R4. We thank the reviewer for highlighting this concern. We fully agree and and changed the title to “Burnout Among Healthcare Workers: Insights for Holistic Well-Being”.

Round 2
Reviewer 3 Report
Comments and Suggestions for Authors
Thank you for the revision and thoughtful response letter. I have no additional comments.